# Approach for Non-Intrusive Detection of the Fit of Orthopaedic Devices Based on Vibrational Data

**DOI:** 10.3390/s23146500

**Published:** 2023-07-18

**Authors:** Constanze Neupetsch, Eric Hensel, Andreas Heinke, Tom Stapf, Nico Stecher, Hagen Malberg, Christoph-Eckhard Heyde, Welf-Guntram Drossel

**Affiliations:** 1Fraunhofer Institute for Machine Tools and Forming Technology, 09126 Chemnitz, Germany; eric.hensel@iwu.fraunhofer.de (E.H.); tom.stapf@iwu.fraunhofer.de (T.S.); welf-guntram.drossel@iwu.fraunhofer.de (W.-G.D.); 2Professorship Adaptronics and Lightweight Design, Faculty of Mechanical Engineering, Chemnitz University of Technology, 09111 Chemnitz, Germany; 3Department of Orthopaedic, Trauma and Plastic Surgery, University of Leipzig Medical Center, 04103 Leipzig, Germany; christoph-eckhard.heyde@medizin.uni-leipzig.de; 4Institute of Biomedical Engineering, Dresden University of Technology, 01307 Dresden, Germany; andreas.heinke@tu-dresden.de (A.H.); nico.stecher@tu-dresden.de (N.S.); hagen.malberg@tu-dresden.de (H.M.)

**Keywords:** socket fit, lower-limb prosthesis, transfemoral amputation, residual limb, local pressure mark, structural dynamics, measurement method, acoustic sensor, frequency response analysis

## Abstract

The soft tissues of residual limb amputees are subject to large volume fluctuations over the course of a day. Volume fluctuations in residual limbs can lead to local pressure marks, causing discomfort, pain and rejection of prostheses. Existing methods for measuring interface stress encounter several limitations. A major problem is that the measurement instrumentation is applied in the sensitive interface between the prosthesis and residual limb. This paper presents the principle investigation of a non-intrusive technique to evaluate the fit of orthopaedic prosthesis sockets in transfemoral amputees based on experimentally obtained vibrational data. The proposed approach is based on changes in the dynamical behaviour detectable at the outer surface of prostheses; thus, the described interface is not affected. Based on the experimental investigations shown and the derived results, it can be concluded that structural dynamic measurements are a promising non-intrusive technique to evaluate the fit of orthopaedic prosthesis sockets in transfemoral amputee patients. The obtained resonance frequency changes of 2% are a good indicator of successful applicabilityas these changes can be detected without the need for complex measurement devices.

## 1. Introduction

The development of comfortable and effective prosthetic sockets for lower-limb amputees remains a challenge in the field of medical engineering. An optimal prosthetic socket must be comfortable for the user and simultaneously ensure stable fitting and proper load transmission in the tissue, especially for prostheses for the lower limbs. The interaction between the socket and residual limb can be crucial to achieve a comfortable and stable fit that reduces the risk of lasting damage to the tissue and the residual limb. While orthopaedic technicians take into account biomechanical parameters, the perception of comfort by patients remains the ultimate factor in determining the success of a prosthetic device. Previous studies have shown that up to 50% of transtibial amputation patients do not use their prostheses regularly due to discomfort associated with the socket. Moreover, the non-use rate is even higher among transfemoral amputation patients. Among those who do not regularly use their prostheses, 35.3% attribute their lack of usage to a lack of comfort [1,2,3]. This highlights the pressing need for improved methods to assess and optimize the fit of prosthetic sockets, aiming to alleviate discomfort and enhance patient satisfaction.

However, the socket geometry only represents a snapshot of the residual limb, and short-term and long-term changes cannot be fully or adequately compensated for. The soft tissues of the residual limb are subject to large volume fluctuations over the course of the day. Negative influences, such as local pressure marks and lower carrying comfort, can result from these volume fluctuations.The pressure distribution is heterogeneous throughout and peak load locations during the gait cycle are highly related to the level of comfort [4,5,6,7]. Recent studies indicate that tissue-specific load distribution reduces the risk of permanent tissue damage to the residual limb. These studies also suggest the potential for automation of the adaptation of prosthetic socket geometry to optimize fitting quality. However, continuous measurement of objective parameters during everyday activities will be necessary to facilitate such automated adaptations [8,9].

Existing methods for measuring interface stress face various limitations, including the need for holes in the socket, the use of bulky sensors that disrupt the environment within the socket, and the presence of cables [10]. Most recent studies present thin pressure and shear sensors that are sheet screen-printed on polyethylenterephthalat (PET) [11]. Similar approaches utilised a low-cost piezoresistive material known as Velostat to integrate strips of piezoresistive sensors mapping the pressure on the residual limb [12,13]. However, placing these sensors between the socket and the skin makes them prone to producing creasing in the PET substrate. To address this, initial proofs-of-concept have demonstrated the seamless integration of a capacitive and resistive shear and force sensor on the silicon liner component of the prosthetic limb [14,15]. Alternatively, 3D-printed liquid metal-based soft-pressure sensors for health-monitoring applications have been utilised for pressure sensing in smart liners [16]. Other recent studies have explored the use of optical systems or an optomechanical sensor principle. The former approach uses fibre Bragg grating [17]. The second uses the optomechanical sensor principle, where two types of fibres, a driving and a sensing fibre, are aligned orthogonally. When a normal force is applied to the crossing of these fibres, light leaks from the driving into the sensing fibre [18]. Both types of sensors can be recoated and embedded in an epoxy material to form a sensing pad and then further embedded in a silicone polymer material to form a pressure sensor. Another approach is inductive sensing to measure residual limb displacements at multiple locations in the prosthetic socket, which may serve as a valuable tool to understand socket fit [19].

To date, none of the sensor systems presented have demonstrated the capability to consistently measure the in-socket pressure of a transfemoral prosthesis over an extended period of daily use. This should include hundreds of thousands of gait cycles. The various sensors presented have limited accuracy and reliability over extended periods of time. Furthermore, the mechanical properties of the socket itself, including its elasticity and deformation characteristics, can impact the precision of pressure measurements. The physiological changes that occur in the residual limb, such as changes in soft tissue volume and muscle activity, can affect the pressure distribution and make it difficult to obtain consistent and reliable measurements. Integrating measurement technology often results in disturbances or significant adjustments to the socket’s topography to accommodate the sensors effectively. This, in turn, has a negative effect on the enclosed tissue.

Given these challenges, indirect sensor systems that can provide accurate and reliable pressure-correlated measurements over extended periods of time might be more suitable for this task. Moreover, there is a need for non-intrusive analysis methods to evaluate pressure distribution in prosthetic sockets. Structural dynamics measurement methods, which have been successfully applied in various fields, including acoustics, represent a promising approach for addressing this issue. However, there is a lack of research regarding the application of structural dynamics analyses to prosthetic sockets and the effect of boundary conditions (BCs) on the accuracy of structure-borne sound measurements using carbon fibre-reinforced plastic.

This study explores the use of structural dynamics measurement methods—specifically, structure-borne sound—as a potential non-intrusive method for analysing local pressure marks in prosthesis sockets. This approach builds on previous research that has demonstrated the transferability of experimental acoustic analysis to medical engineering problems and the acquisition of modal parameters for parameter identification [20,21]. This novel approach comes with new challenges regarding the development of an experimental setup that can offer sufficient transferability to the real prosthesis but allows for enough flexibility to incorporate the latest findings into the configuration of the measurement setup.

To address this gap in knowledge, this study aimed to investigate the feasibility of using structure-borne sound measurements to analyse the pressure distribution in prosthetic sockets with varying BCs. The results provide an important basis for the development of a new measurement method allowing the identification of changes in local pressure marks during the wearing of a prosthesis. This should help to improve the fitting and comfort of transfemoral prostheses and ultimately enhance quality of life for amputees.

## 2. Materials and Methods

### 2.1. Object of Investigation

The test object used in the experiment was an anatomically shaped socket for a transfemoral prosthesis manufactured out of carbon fibre according to current standards. Manufacturing details are reported in [20]. A pressurised air pocket mimicked the residual limb and its volume fluctuation during the preloading of the socket. To ensure reproducible loading conditions for each trial, a standardised medical-grade billary drainage bag with 1500 mL volume was used (ASID BONZ GmbH, Herrenberg, Germany) as an air-tight volume. Each drainage bag (henceforth referred to as a bag) was inflated with an air pressure of 500 mbar to stretch the material and check for any leakage prior the experiment. A custom-designed pressure regulator was used in the experimental setup. Proportional pressure regulators and valves from Festo (Festo Vertrieb GmbH and Co. KG, Esslingen, Germany; *type* MS6-LRP-3/8-D4-A8 and MA-50-16-1/4) controlled the airflow into and out of the bag. A digital pressure gauge (PCE Instruments GmBH, Meschede, *type* PCE-DPG 25) measured and displayed the output pressure with an accuracy of 0.25% of the scale. During structural dynamics measurements, the bag was placed inside the socket touching the bottom, with the flat side orientated toward sensor locations one and three (Figure 1). The bag was inflated in a controlled manner to simulate different volumetric conditions.

### 2.2. Experimental Setup

The aim of the investigation was the identification of changes in the dynamical behaviour of a prosthesis due to pressure changes at the interface between the residual limb and prosthesis socket. The dynamical behaviour of the structures was analysed using structural dynamic measurements and analysis techniques to evaluate vibrations of the system caused by internal or external excitations.

Additional pressure sensors were attached inside the prosthetic socket to serve as a reference for monitoring the internal load state of the prosthetic socket. Currently, the most widely utilised commercial sensing system for performing pressure measurements in prosthetics in a clinical setting is the piezo-resistive F-socket system. To minimize the impact on the vibration characteristics of the prosthesis, we opted for the smallest and lightest sensors possible. We used FlexiForce sensors (Tekscan Inc., Boston, MA, USA, *type* FlexiForce sensor A201) with a force range of 111 N and a sensor length of 190.5 mm because of their well-studied low measurement error and minimal sensor drift for pressures under 100 kPa (1000 mbar) [22]. The sensor is constructed of two layers of flexible polyester film substrate, each coated with a layer of conductive silver ink. These two layers are then joined together using a thin, compressible polymer material. When pressure is applied to the sensing area, the polymer material is compressed, resulting in an increase in the conductivity of the polymer and, consequently, an increase in the overall conductivity of the sensor. The sensor has a thickness of 0.2 mm and a sensing area diameter of 9.53 mm. The four sensors were placed evenly spaced around the socket to measure the load discrepancies in the frontal and sagittal plane. Wax along the sensor leads fixed them on the socket. Similar sensor placement can be found in [23,24]. These zones were anatomically associated with the most pressure-tolerant and pressure-sensitive regions. The first extended from the large abductors and across the thigh and from the gluteal muscle to the lateral hamstrings. The second lay medially proximal near the ischium [6]. Within the experimental investigations, the FlexiForce sensors were used to acquire contact forces between the bag and prosthesis (cf. Figure 2).

Since a separated prosthesis is, from a mechanical point of view, a passive system without an internal excitation mechanism, utilisation of external excitation is required. In Figure 2, a schematic representation of the used test setup is shown.

Within the current investigations, the external excitation of the prosthesis was realised by an electrodynamic shaker (Brüel & Kjær, Nærum, Denmark, Mini-shaker *type* 4810). The shaker’s passive side was supported by elastic springs within a frame and attached to the prosthesis by a two-component adhesive. In order to minimise the effect of unwanted transversal forces, a so-called stinger (metal rod with a small diameter) was used as a connector between the shaker and the test structure. The shaker was operated by a signal waveform generator (HIOKI E.E. CORPORATION, Ueda, Japan, waveform generator *type* 7075) whose output signal was amplified by supplementary hardware (Brüel & Kjær.: power amplifier *type* 2716-C). An important dynamical quantity in a mechanical system is the input impedance relating the force and the velocity at the excitation location. Thus, a dynamic impedance sensor (PCB Piezotronics, Inc., Depew, NY, USA, *type* 288D01) was applied at the excitation location. The impedance sensor was directly attached to the prosthesis and the stinger of the used shaker. In addition, a triaxial accelerometer (PCB Piezotronics, Inc.: *type* 356A45) was attached on the opposite side of the prosthesis. As a data acquisition system, a PAK MKII (Müller-BBM VibroAkustik Systeme GmbH, Planegg, Germany, *type* PAK MKII) was used.

In general, test structures are analysed under idealised BCs during structural dynamic measurements; i.e., the structure is either freely supported inside a frame or fixed to the ground. In both cases, the assumed idealised BCs need to be ensured for the entire frequency range of interest. In terms of a fixed support, one has to ensure that the base structure where the test object is mounted provides a sufficiently high impedance at the connection point. Usually, this is satisfied for lower frequencies below the first eigenfrequency of the base structure. On the other hand, a free support, usually realised using soft springs, needs to be adjusted to the eigenfrequencies of the test structure. An ideal freely supported system exhibits rigid body modes at 0 Hz. In practice, this ideal support cannot be realised since the non-zero stiffness of the springs leads to rigid body modes > 0 Hz. For free supports, one has to ensure that the rigid body modes are lower than 10 to 20% of the first flexural mode (cf. [25]). In the case of the analysed prosthesis, a free suspension was chosen using elastic straps. Rigid body modes were found at frequencies below 20 Hz and the first flexural mode of the prosthesis was determined at approx. 500 Hz, indicating that the described condition was satisfied. The left part of Figure 1 shows the described test setup for structural dynamic measurements.

The test setup aimed for maximised decoupling by only connecting the bag to the pneumatic load unit and the force sensors to the amplifier. To avoid vibration transfer from the load unit to the prosthesis, the hose between the two was closed after reaching the target pressure. Thus, the pneumatic load unit became inactive.

For structural dynamic measurements, the shaker was operated using a random signal waveform with band-limited noise in a frequency range from 0 to 3000 Hz. For all measurements, force and acceleration signals were acquired for 30 s. Within the following descriptions, the measurements with the shaker and impedance sensor are referred to as frequency response function (FRF) measurements.

In addition to the measurements described above, an experimental modal analysis (EMA) was carried out using a 3D Scanning Laser Doppler Vibrometer (Polytec GmbH, Waldbronn, Germany, *type* PSV 400). The objective of these additional measurements was to gain information about the mode shapes in the considered frequency range up to 3 kHz and correlate global and local mode shapes to the results of FRF measurements. EMA was carried out using the same BCs as specified in the previous section. For the sake of simplicity, the excitation position of the shaker remained the same. In order to gain optimal signal quality for the optical measurements, the surface of the prosthesis was prepared using retro-reflective tape. For the vibrometer measurements, the test structure was excited using a pseudo-random signal waveform and the system’s responses were acquired at 105 scan points. The large number of response nodes enabled a detailed representation of global and local mode shapes where dominant displacements could be concentrated in a small area of the investigated test structure. In contrast to the FRF measurements, EMA was carried out without force sensors and bags to minimize the influence on the prosthesis’ dynamical behaviour.

### 2.3. Test Program and Signal Processing

FRF measurements were carried out with one prosthesis with three different bags (identical in construction) to observe the reproducibility of the utilised experimental test setup and measurement technique. After placement of the static force sensors and insertion of the bag, the bag was pre-strained with a system pressure of 500 mbar applied with a pneumatic pressure gauge to ensure the correct position of measurement components. Subsequently, FRF measurements were carried out for different system pressures at 100 mbar, 250 mbar, 500 mbar and 750 mbar with the measurement settings described in Section 2.2.

In subsequent data post-processing, the acquired time data for the impedance sensor (force and acceleration) were transformed into the frequency domain to relate the response acceleration to the exciting force as inertance *A*
(1)$$\begin{array}{c} A\left(\omega \right)=\frac{a(\omega )}{F(\omega )} \end{array}$$
where A(ω) was calculated as the frequency response function H1 (cf. [25]) using 25 averages and a frequency resolution of 500 mHz. In contrast, the time signals of the attached force sensors were not converted into the frequency domain since they only included a static force that remained constant for a certain system pressure for the duration of one single measurement. For clarification, the individual steps of data processing are shown schematically in Figure 3.

As mentioned above, the acquired FRFs were calculated using a frequency resolution of 500 mHz; i.e., FRF values were only obtained with discrete frequency steps. The changes in the dynamical behaviour of the prosthesis due to varying system pressures were evaluated based on shifts in resonance frequencies. Due to the frequency discretisation, the local extrema of the FRFs could be found between two frequency steps, which meant that the frequency values where resonances occurred could be inaccurate. In order to overcome this problem, the acquired FRFs were analysed using a curve fit within the frequency range around a certain resonance. The basis of the curve fit algorithm used here was the notation for an FRF in pole–residue format (cf. [26])
(2)$$H\left(\omega \right)=\frac{r}{j\omega -\lambda }+\frac{r^{*}}{j\omega -\lambda^{*}}$$
where *r* denotes the residue and λ denotes the pole of the FRF *H* in a certain frequency range. Based on Equation (Equation 2), a least squares curve fit was performed to obtain *r* and λ. In a subsequent step, the FRF was synthesised using the obtained residue and pole to determine the frequency of the FRF’s maximum according to the following equation
(3)ωmax=argmax|ImHsynt(ω)|.

In Equation (Equation 3), the imaginary part of the FRF is used. As described above, the acquired FRF has the form of inertance, relating acceleration to force. For inertances, the imaginary part is directly related to the motion amplitude of a structure. In contrast, the real part of the inertances is characterised by a root. Thus, the real part of inertance can also be utilised to determine the frequency of an FRF maximum, but a different approach needs to be chosen instead of Equation (Equation 3). Figure 4 contains a corresponding example to clarify the use of the described curve fit. The left part of Figure 4 contains a measured FRF in a chosen frequency range (black solid line), the results of the curve fit at certain frequencies (white dots), and the frequency range used to perform the curve fit (blue highlighted area). In addition, the right part of the figure contains the synthesised FRF, which can be utilised afterwards to identify the frequency of the FRF’s maximum according to Equation (Equation 3).

## 3. Results

### 3.1. Force Measurements

When the bag expands due to applied pneumatic system pressure and is in contact with the prosthesis, a static force can be detected at the interface between the bag and the prosthesis. The amplitude of this static force is directly dependent on the pressure in the bag. Thus, it is expected that an increasing system pressure will lead to an increase in the measured force.

In the first step, the force sensors and the corresponding measurement chain consisting of the measurement bridge and hardware low-pass filter were calibrated using weights with precisely determined masses (nine different weights with masses in the range from 9 g to 390 g). The results of the calibration (units: V/N) are shown in Figure 5. Since a linear relation between the applied force and the resulting voltage was expected, a subsequent linear regression was carried out (represented as solid lines in Figure 5) in order to allow interpolation and extrapolation of the measured values.

Based on the test program described in Section 2.3, four static forces were acquired for three different bags each with 100 mbar, 250 mbar, 500 mbar and 750 mbar internal pressure imitating the volume fluctuations in the amputated residual limb provided by the utilised pneumatic load unit. The numeration of forces corresponds to the locations in Figure 1 (right part). The acquired static forces for all measurements at all locations are presented in Figure 6.

### 3.2. Resonance Frequencies

In Section 3.1, the changes in contact forces due to varying system pressure inside the bag were presented and discussed. It was shown that, using the chosen test setup, a detectable change in the contact between the bag and prosthesis could be realized. Based on these results, the next part of the article presents the results of the dynamic FRF measurements. Before discussing the results for the changes in the dynamical behaviour for different system pressures, a brief evaluation of the quality of the obtained data should be presented.

Figure 7 shows an acquired input inertance and the corresponding coherence at the left side, as well as the measured input force at the right side, in a frequency range from 0 Hz to 3 kHz. The inertance FRF shows major resonance peaks in the considered frequency range, indicating the presence of eigenfrequencies in the investigated system. It should be noted that, since no modal parameter extraction was carried out with the acquired data, these resonance frequencies are not labelled as eigenfrequencies within the following descriptions. In addition to the resonance peaks, a major antiresonance could be found at approx. 350 kHz. This antiresonance was caused by a very low acceleration since the corresponding force at this frequency exhibited no significant drops (compare the right part of Figure 7). The antiresonance shows that the prosthesis exhibited (nearly) no movement at the excitation location within this frequency range. The presence of this antiresonance caused a drop in the coherence at approx. 350 Hz (cf. [25]). In general, the high coherence indicates valid measurements within the considered frequency range up to 3 kHz.

The right diagram in Figure 7 contains the excitation spectrum. It shows that—for the chosen frequency resolution of 500 mHz—a maximum force of 13 mN could be found at approx. 2 kHz, and in the frequency range up to 1 kHz, an average input force of approx. 6 mN was measured. The changes in the excitation forces, especially for frequencies above 1 kHz, were caused by the dynamical behaviour of the prosthesis and its reaction to the impedance sensor. The acquired dynamic forces were low in comparison to the static forces, indicating that the interface between the bag and the prosthesis would not be influenced by an external excitation. Since the input force and the obtained input inertance were mainly characterised by the properties of the prosthesis itself, no major differences were found between the various measurements. Thus, no further evaluations or comparisons for different measurements regarding coherences and input forces were required.

Following the test program described in Section 2.3, a total number of 12 input inertances were acquired. A first rough overview of the determined FRF revealed that varying system pressure did not affect any of the resonance peaks, which will be discussed later based on the results of the experimental modal analysis. Therefore, the following resonance frequencies revealing a dependency on the applied system pressure were found:$$f=\left[500,\;560,\;700,\;1660,\;2675\right]\;\;\mathrm{Hz}$$

The resonances around these frequencies were analysed in detail using Equation (Equation 3) based on thecurve fit approach described in Section 2.3. The curve fit was carried out for each measurement, leading to an overall value of 60 frequencies (five resonance locations for 12 measurements). Figure 8 contains the results for the obtained resonance frequencies for all measurements.

Each subplot in Figure 8 shows the changes in certain frequencies due to changes in system pressure. First of all, it is obvious that resonance frequencies increased with increasing system pressure for pressure values ≥ 250 mbar. Especially for the lowest pressure of 100 mbar, the frequencies at 560 Hz, 1660 Hz, and 2675 Hz seem implausible due to the higher values in comparison to the measurement at 250 mbar. A closer look at the data for 100 mbar and 250 mbar shows that this behaviour cannot be recognised for all investigated variants. For example, the data for bag three show monotonically increasing frequency values at 2675 Hz. In contrast, the data acquired for bags one and two exhibited lower frequency values for 250 mbar in comparison to frequencies obtained at 100 mbar system pressure. Since this behaviour was not observable for all measurements, it can be noted that certain resonances were more sensitive to low contact pressures. This behaviour may have been caused by slight misalignments of the bag not compensated for at low system pressures. In addition, Figure 8 shows that major differences for bag one could be found for resonance frequencies at 560 Hz and 700 Hz, which was obviously caused by a change in the initial conditions of the investigated system consisting of the prosthesis and the bag. This change in the initial conditions could have been caused by misalignment due to manual positioning of the bags. Nevertheless, increasing frequencies with increasing pressure can also be recognised for bag one.

In order to gain comparability despite the varying initial conditions, the change in a certain resonance frequency $\Delta f_{i}$ can be expressed in a relative way using the following equation:(4)$$\Delta f_{i}=\left(\frac{f_{i}}{f_{p=100\mathrm{mbar}}}-1\right)\;\cdot \;100\%$$
with reference frequency $f_{p=100\mathrm{mbar}}$ at 100 mbar. Applying Equation (Equation 4) to the obtained data shown in Figure 8 led to a representation of relative frequency changes referenced to the initial condition at 100 mbar, as displayed in Figure 9.

The applied relative scaling showed that initial condition variations, especially those observed for 560 Hz and 700 Hz, did not affect the relative changes in resonance frequencies. In addition, the comparison showed that relative changes in frequencies were not constant for the investigated resonances. Maximum relative changes could be found in a range from 0.2% to 2.1%. In particular, the comparison of the changes at 500 Hz and 560 Hz clarified that different resonances were affected in different ways by the interface of the prosthesis and bag imitating the residual limb. The absolute changes in frequencies could be found in the range from 3 Hz to 10 Hz for all resonances.

In order to provide information about the relation between resonance frequency changes and changes in obtained contact forces, the relative frequency changes dependent on the acquired forces at sensor locations one and four are shown in Figure 10.

The relative changes in resonance frequencies shown in Figure 10 were calculated according to Equation (Equation 4) and, thus, they were comparable to the changes presented in Figure 9. Based on the force measurement results at different sensor locations (cf. Section 3.1), the data for sensor locations two and three were omitted due to possible misalignment or unsuitable placement. Nevertheless, the data shown in Figure 10 reveal that even small changes in contact pressure lead to a noticeable shift in resonance frequencies. This can be seen in particular for resonance frequencies at 500 Hz and 700 Hz, where relative changes larger than 1.5% could be identified. For the resonance frequency at 500 Hz especially, an increase in contact force from 5 N to 10 N led to a relative frequency change of approx. 1%.

### 3.3. Experimental Modal Analysis

In order to gain further information about the dynamical behaviour of the prosthesis, an additional experimental modal analysis (EMA) was carried out. The objective of an EMA is to identify the natural vibration behaviour of a structure through the determination of eigenfrequencies, mode shapes, and modal damping values. As described in Section 2.2, EMA was carried out using a free support and shaker excitation, which was identical to the setup for the FRF measurements. In contrast to the FRF measurements, the bag and the static force sensors were removed and a 3D scanning laser Doppler vibrometer was used for acquisition of the system’s responses due to the external excitation realised by the electrodynamic shaker. Another difference in comparison to the FRF measurements was the position where the prosthesis was mounted. In order to obtain a scan area as large as possible, the prosthesis was supported freely at the distal end. Since the free support decouples the structure at very low frequencies (compared to the expected eigenfrequency range of >500 Hz), the influence of the support on the dynamical behaviour of the test object was expected to be small. In this way, the test setup allowed the definition of a fine response node mesh of 105 scan points. The use of a fine mesh enabled the detection of modes with local mode shapes; i.e., mode shapes with dominant amplitudes at a limited spatial area.

Within the EMA measurements, transfer admittance FRFs between excitation and response nodes were obtained (admittances due to the acquisition of velocities). These admittance FRFs were utilised in a subsequent step to perform a so-called modal curve fitting in order to obtain the modal parameters: eigenfrequencies, mode shapes, and modal damping values. EMA results revealed that the investigated prosthesis held 10 flexural modes in the frequency range up to 3 kHz, while rigid body modes at low frequencies were not considered. The first mode was found at 501 Hz and the eigenfrequencies correlated with the resonance frequencies obtained during FRF measurements presented in Section 3.2. This correlation was expected and clarified that the resonance frequencies were caused by the dynamical properties of the investigated prosthesis. This is an important fact since it can be concluded that the resonance peaks were not caused by supplementary devices, like force sensors or the bag. However, these devices can lead to small deviations; e.g., the eigenfrequency at 709 Hz obtained in the EMA and the corresponding resonance frequency at 700 Hz obtained in the FRF measurements.

Figure 11 presents a chosen set of mode shapes in order to gain comparability with the FRF measurement results presented in Section 3.2. The displayed mode shapes were dominated by major deflections at the proximal end of the prosthesis. This was plausible due to the geometrical shape of the structure, which had an open end in the proximal direction and a closed end in the distal direction. In addition, one can observe that mode shapes had a global character for lower frequencies, which means that larger areas contributed to the corresponding shape. In contrast, mode shapes became more local in higher frequency ranges; for example, the mode shape at 1691 Hz. Figure 11 contains three mode shapes (at 501 Hz, 709 Hz, and 1691 Hz) for which changes in FRF resonances were detected due to variation of the system pressure (compare Figure 8 and Figure 9). In contrast, for the mode shape obtained at 1380 Hz, no distinct changes in FRF resonances were observed. Based on the chosen test setup and the related locations for the contact between the bag and prosthesis, it is noticeable that mode shapes were not influenced by pressure changes, and the blue-coloured area of the example mode shape at 1380 Hz shown in Figure 11 had comparable low deflections for a certain mode shape. The described area was the edge created during the modelling of the prosthesis shape between the medial and anterior surfaces of the prosthesis socket.

## 4. Discussion

### 4.1. Acquired Forces Dependent on Applied System Pressure

The sensors inserted in the interface were suitable for the experimental setup and for developing the measurement method. This ensured that the pressure fluctuations imitated by the bag also resulted in a change in the forces applied to the prosthesis.

The measured data show that force–pressure dependencies were quite similar for all three bags at sensor locations one, two and four. The value for 100 mbar at location two was close to zero, indicating that the bag did not have sufficient contact with the prosthesis during this measurement. In addition, the force value for bag one seemed to be too small and the value for bag three appeared to be too large at this position for the considered pressure of 100 mbar. This indicated that the chosen sensor location two seemed to be unsuitable for low pressures since the contact between bag and prosthesis could not be reproduced in an acceptable way. For the remaining system pressures at force sensor location two, the measured forces increased with increasing pressures but, in comparison to locations one and four, with comparable low values. The maximum contact force at this position could be found at approx. 1N at the maximum applied system pressure of 750 mbar. In contrast, the maximum forces for sensor locations one and four were found in the range between 11.4 N and 14 N for the highest system pressure applied to the system. The comparable low static force values acquired at sensor location two can be explained by the non-ideal location of the sensor in relation to the interface between the bag and prosthesis, as the area of the maximum contact force may not have been covered by the location of the sensor. Nevertheless, and despite the deviations at sensor location two for the lowest system pressure at 100 mbar, the acquired force values were reproducible for force sensor positions one, two, and four. The comparison of the results obtained at sensor location four revealed an offset of approx. 1 N for bag one. This offset could have been caused by a misalignment of the bag inside the prosthesis for this sensor location since the bags were positioned by hand without utilisation of any fixture or alignment device. Comparing the dependency of the measured forces on the applied system pressure for sensor locations one and four, it is obvious that they showed asymptotic behaviour. This behaviour was plausible since it was expected that the contact pressure would be distributed over a larger area for higher pressure due to the elasticity of the bags, leading to the observed asymptotic curve shape for the contact forces. Finally, the force values acquired at sensor location three were not plausible since they indicated a nearly constant dependency for the contact forces on the system pressure for values > 100 mbar. This behaviour can be explained by the position of the sensor inside the prosthesis, which was located in the area of a geometrical undercut. Due to this undercut, the position of the force sensor seems to have been undetermined and, especially for larger pressures, contact behaviour remained unchanged.

The sensors induced a topological modification in the sensitive interface between the amputated residual limb and the prosthesis. This impact on the interface may lead to negative effects on the interface, as described in the introduction of the paper. The force sensors in the sensitive interface are only needed in the development phase of the presented method. After the successful development of the structural dynamic method for the detection of volume fluctuations, they are no longer required for use with patients. The redundancy of the sensors inside the prosthesis socket clearly distinguishes this research approach from others [27,28]. Other sensor technologies require direct skin contact [12,13]. Non-physiological stress in this interface often leads to tissue compression, with corresponding negative consequences [29]. In addition, patients often suffer from comorbidities, such as diabetes and vascular diseases, with sweating abnormalities and a reduced convective mechanism in the circulatory system [30], which can lead to sensor slippage. The lack of comfort can be another major obstacle for the user of the prosthesis [1,2,3]. In particular, the complex process of putting on and removing the prosthesis is critical [31]. With the presented method, there are no restrictions due to the position of the sensors. This promises considerable added value compared to the state-of-the-art technologies.

### 4.2. Shift in Resonance Frequencies

The presented results show that the obtained resonance frequencies varied for the three bags used within the experiments. However, this is not a limitation for later use of the method with patients since the objective of this method is to detect changes caused by pressure variations in the prostheses. As shown in this paper, variations in the structural dynamics of prostheses can be detected if pressure at certain locations changes. It is assumed that the presented method is adaptable to patients. It can be expected that the patient will experience the lowest pressure at the beginning of the day and, in the course of the day, higher pressures could be effective due to physiological processes. Everyday activities, such as putting on and removing the prosthesis or physical activities, have a direct effect on the volume of the amputation residual limb. The expected values also depend on the time of day and the time interval following the activity (stabilisation time) [9]. Hence, dynamic quantities can be utilised to identify these changes by comparing measured data to an initial state (e.g., obtained at the beginning of the day). For resonance frequencies at 560 Hz and 700 Hz, relative changes in the range of 1% to 2% were identified. It should be mentioned that the investigated prosthesis had greater thickness in comparison to conventional prostheses. Therefore, the utilised prosthesis held a comparable high local stiffnesses and, thus, the impedance ratio between the prosthesis and the tissue was higher compared to conventional prostheses with smaller thicknesses. Due to this, it was expected that the relative frequency changes for conventional prostheses would be larger in comparison to the changes obtained. Both the results and basic structural dynamics considerations show that, in all investigations based on structure-borne sound, the prevailing boundary conditions were essential [32]. However, investigations with structures made of carbon fibre-reinforced plastic following the geometric model for prosthesis sockets for a system characterisation with different BCs using structure-borne sound measurements are not present in the literature.

### 4.3. Experimental Modal Analysis

Of course, the correlation shown between the mode shapes and changes in the acquired FRFs due to pressure variations is only valid for the chosen test setup and, thus, for the locations of the contact areas between the bag and prosthesis. If these positions change, an effect on other mode shapes and corresponding FRF resonance frequencies can be expected. Due to the high inter- and intravariability of prosthetic sockets, this will also be the case for any other form of characterisation. Any measurement on the patient is only a snapshot of the amputation stump, as it is already known that soft tissue has highly nonlinear and time-dependent behaviour [28,29,33]. Nevertheless, the correlation shown between the mode shapes and changes in FRFs clarifies that local contact areas influence the dynamical behaviour of prostheses. In addition, and based on the results of the EMA and FRF measurements, it is expected that the position of local contact areas can be identified by analysing the dynamical behaviour.

### 4.4. Bags

The bag used allowed the possibility of exposure to varying pressures; thus, it was suitable for the experimental setup. The pressure applied vertically to the contact surface between the inner socket wall and the bag was considered equal to the air pressure applied. A similar approach was successfully implemented by [11]. In contrast, no metal encasement was used to avoid any parasitical effects on the dynamical behaviour of the socket. The frictional force between the bag and the inner socket wall was sufficient to prevent the displacement during the experiment and unpredictable pressure conditions in the socket. A reasonable range for an amputee with a body mass of 80 kg during daily life activities like walking can be assumed to be a minimum of approx. 10 kPa (100 mbar) for the anterior proximal position (sensor location three) to 60 kPa (600 mbar) for the posterior proximal position (sensor location one). These values are also lower than the reported pain-raising pressure thresholds of 350 kPa (3500 mbar) [34]. The pressure range can be divided into several experimental studies with determined boundaries during walking [7,35,36,37,38]. It is hence possible to imitate the volume fluctuations acting in the amputated residual limb with technical aids, which is essential for the current analyses. It should be kept in mind that the presented results and changes in dynamical behaviour were only obtained for one exemplary position and, thus, one exemplary inner pressure state. In order to gain more flexibility, the utilised bags will be replaced in future research by other materials/structures (for example, silicone) allowing the application of various pressure positions.

## 5. Conclusions

The presented paper describes a basic investigation into a non-intrusive technique to evaluate the fit of orthopaedic prosthesis sockets of transfemoral amputee patients based on experimentally obtained vibrational data. In contrast to the current state-of-the-art, where sensors are applied in the sensitive interface between a prosthesis and the residual limb, the approach shown here is based on changes in dynamical behaviour detectable at the outer surface of the prostheses and, thus, the described interface is not affected. Changes in the fit of the investigated prosthesis were imitated by a pressure bag positioned inside the structure. The bag was connected to a pneumatic auxiliary system to apply an inner pressure. The pneumatic load inside the bag was used to realise the contact between the prosthesis socket and the bag. In order to gain information about the interface and to ensure reproducibility, four force sensors were attached at different locations at the inner surface of the investigated prosthesis socket. The measured contact was used to check if increasing system pressures led to higher forces at the interface. Thus, the sensors were only used to ensure detection of the changes in the interface and that the resulting forces were in a adequate range. The force sensors in the sensitive interface can only be used in non-clinical test environments. During further development of the presented structural dynamic analysis technique, as well as for future tests with patients, the interface force sensors will be omitted. Furthermore, it can be assumed that locally different volume changes occur at the residual limb. However, the simplified version of a one-chamber bag cannot reproduce locally varying conditions. In the continuation of the research work, a substitute model of the residual limb should be built to represent this. Nevertheless, the utilised bag was a sufficient approach to evaluate the applicability of structural dynamic measurements for detecting changes in the fit of orthopaedic prosthesis sockets. Within the experiments carried out, maximum interface forces of 14 N were obtained.

The measurements were carried out for four different system pressures. In addition, the reproducibility was checked by repeating measurements for three different bags. For each pressure state, the system was excited by an electrodynamic shaker connected with a stinger to an impedance sensor attached to the outer surface of the prosthesis socket. An additional triaxial accelerometer was applied at the opposite side of the prosthesis to acquire the system’s response, not only at the driving point where the shaker was connected. In a first step, the acquired frequency response functions were post-processed by curve fitting within the ranges of the resonance frequencies. The application of curve fitting was used to overcome the finite frequency resolution resulting from a finite measurement duration to obtain more accurate locations for the maximum frequencies. The comparison of resonance frequencies for different system pressures revealed that frequency shifts could be detected for different pressure states. It was assumed that the contact area between the bag and the prosthesis socket was not distributed homogeneously and, thus, locations with higher contact forces were found, which was confirmed by the static force measurements. For higher system pressures, the contact areas at the main pressure locations increased due to the elasticity of the used bags. As a consequence of this, the local stiffness distribution in the investigated system changed and the corresponding effects were identified by the structural dynamic measurements carried out. For increasing system pressures, a shift in resonance frequencies to higher values was observed, which was plausible due to the enlargement of the contact areas. For different resonance frequencies, a maximum relative change (in relation to the determined frequency at the lowest system pressure) of approx. 2% was found. It should be noted that different resonance frequencies were affected in different ways and the lowest changes were found to be approx. 0.2%. Due to the chosen test setup, the acquired resonance frequencies correlated with the eigenfrequencies of the investigated prosthesis. Therefore, an additional experimental modal analysis was carried out. The objective of the modal analysis was to gain information about the mode shape occurring at a certain frequency. The evaluation of the modal analysis clarified that the obtained differences in the frequency changes were plausible due to the varying contributions of different areas for different mode shapes. The results of the paper can be summarised in brief as follows: Changes in local pressure mark locations between residual limbs and prosthesis sockets lead to a change in dynamical behaviour (shift in resonance frequencies);System changes in dynamical behaviour can be detected by non-intrusive dynamical measurement at the outer surface of the prosthesis socket;The shift in the resonance frequencies depends on the contact force, with increasing forces leading to increasing frequencies;Changes in resonance frequencies correlate with the positions of local pressure marks and the dominant surfaces of the corresponding mode shapes.

Based on the experimental investigations shown and the derived results, it can be concluded that structural dynamic measurements are a promising non-intrusive technique to evaluate the fit of orthopaedic prosthesis sockets in transfemoral amputee patients. The obtained resonance frequency changes of 2% were a good indicator of successful applicability since changes can be detected without using complex measurement devices. Nevertheless, the changes shown due to variations in the system pressures were only investigated for one chosen position for the bag used and, thus, for only one set of contact areas imitating local pressure marks. In future research work, a more flexible system for pneumatic load application will be developed. This will enable the investigation of resulting changes for other positions of local pressure marks. In addition, the spatial resolution required to detect the location of local pressure areas needs to be analysed in detail.

## Figures and Tables

**Figure 1 sensors-23-06500-f001:**
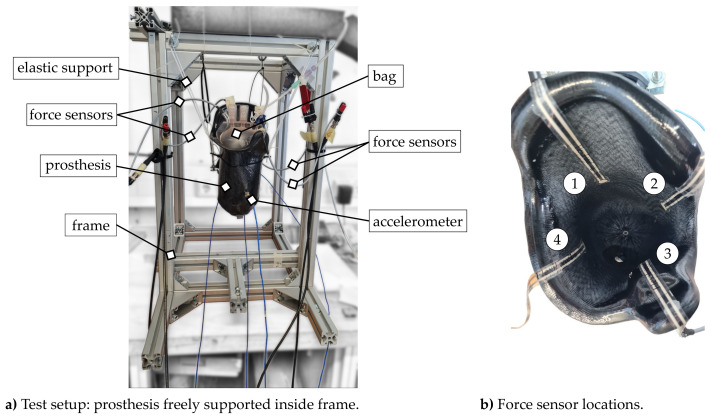
Test setup for experimental analysis of the prosthesis socket.

**Figure 2 sensors-23-06500-f002:**
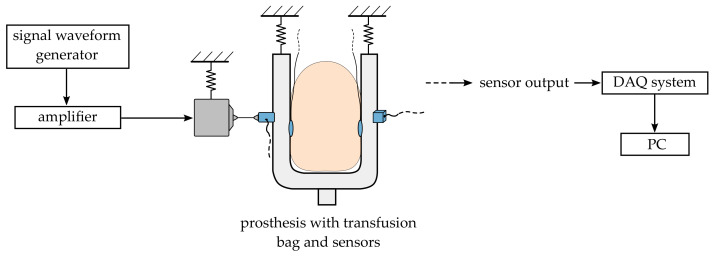
Schematic representation of the test setup.

**Figure 3 sensors-23-06500-f003:**
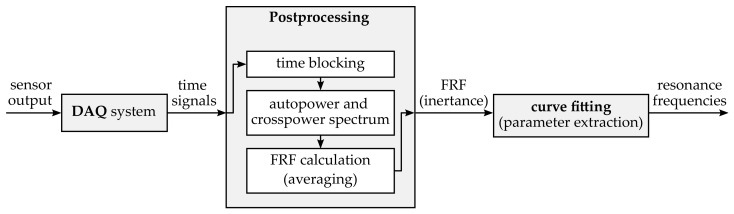
Data acquisition and postprocessing scheme.

**Figure 4 sensors-23-06500-f004:**
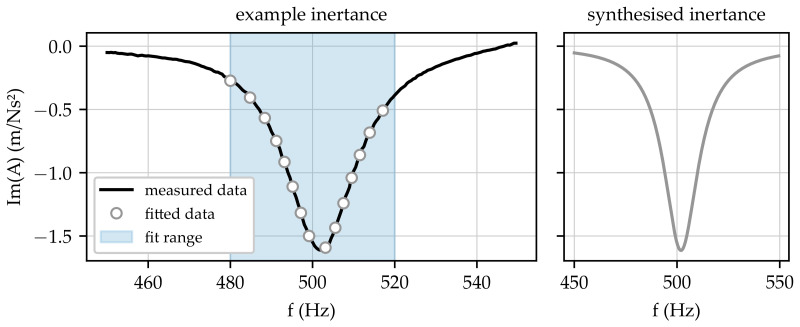
Example of inertance FRF and corresponding curve fit result.

**Figure 5 sensors-23-06500-f005:**
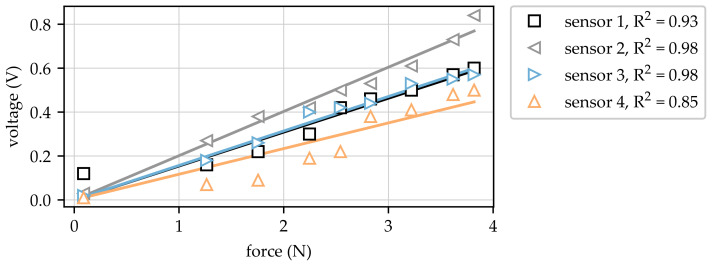
Force sensor calibration and linear regression.

**Figure 6 sensors-23-06500-f006:**
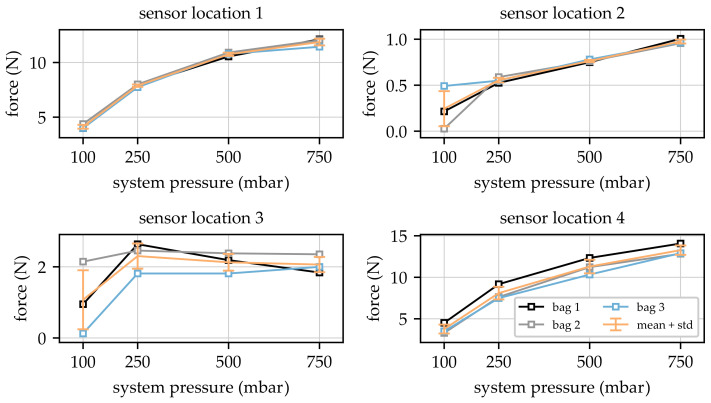
Overview of measured forces for all bags and system pressures.

**Figure 7 sensors-23-06500-f007:**
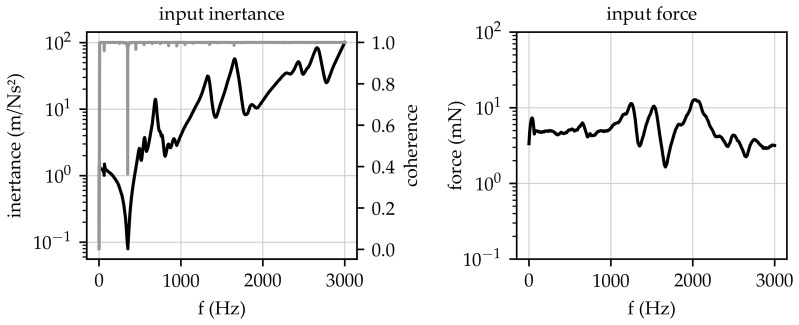
(**Left**) Input inertance (black line) and coherence function (grey line), (**right**) excitation spectrum.

**Figure 8 sensors-23-06500-f008:**
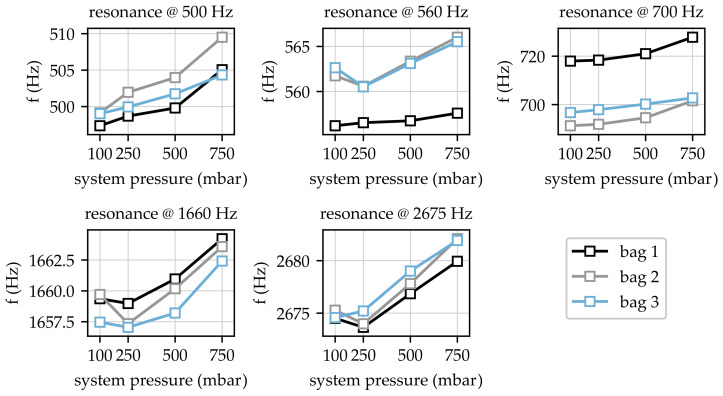
Absolute changes in resonance frequencies due to variations in system pressure.

**Figure 9 sensors-23-06500-f009:**
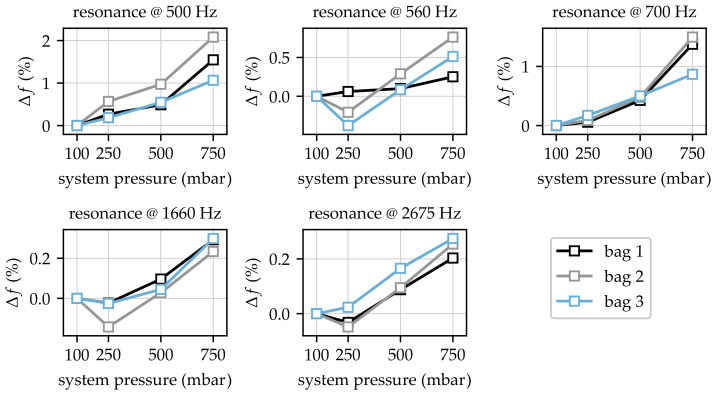
Relative changes in resonance frequencies due to variations in system pressure.

**Figure 10 sensors-23-06500-f010:**
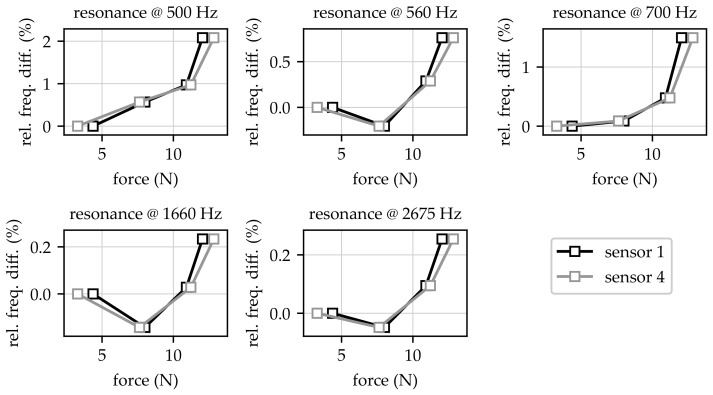
Relative changes in resonance frequencies in relation to acquired contact forces at sensor locations one and four (for bag two as an example).

**Figure 11 sensors-23-06500-f011:**
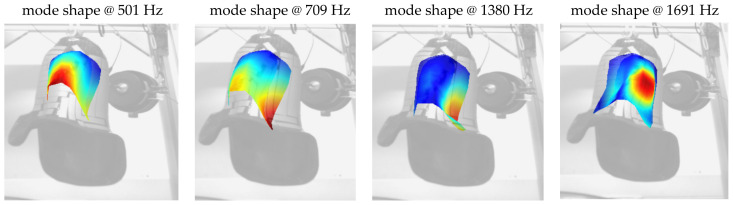
Example mode shapes obtained with experimental modal analysis.

## Data Availability

The data presented in this study are available on request from the corresponding author.

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
