# Peer review of "Approach for Non-Intrusive Detection of the Fit of Orthopaedic Devices Based on Vibrational Data"

_sensors, 2023, doi:10.3390/s23146500_

Round 1

Reviewer 1 Report

The work is interesting as a new methodology and can open up new research scenarios. However, in my opinion, the application aspects need to be considered more.

First of all, it is necessary to better clarify the relationship between the study in progress and the benefits for the patient; i.e. how, having data on how external stresses can increase comfort for patients; how socket characteristics can be changed such as compensating for volume changes or problems.

Liners are usually used in the prosthetic technique; soft interfaces in silicone materials that are worn over the stump.

A more accurate simulation should consider the presence of these interfaces and it should be indicated how with a measurement of force/pressure/humidity/temperature parameters it is able to modify the socket structure.

The bag that simulates the stump is too different from the consistency of the stump and therefore all the data collected is of little use. The use of silicone materials can certainly improve the simulation even if it does not consider the great variety of patients.

It is not clear how the external stimulus is applied, if possible provide a photo or video.

Author Response

Thank you for your comments! You can find the answer in the PDF. 

Reviewer 2 Report

1. In Figure 6, the coherence function image disappears after the resonant frequency reaches 3000hz. Does it mean that it cannot be effectively measured when the resonant frequency is higher than 3000hz or that the experiment only tests the resonant frequency below 3000hz?

2. Is there any reason for selecting the four static force sizes of 100mbar, 250mbar, 500mbar and 750mbar for the three bags in the experiment?

3. In Lines 132, how should the sensors mentioned here be placed? Have you considered whether they should be placed in the sensitive area of the patient corresponding to the receiving cavity?

4. Three different bags with the same structure. According to the image, sometimes bag2 and bag3 display similar rules (560hz and 700hz), sometimes bag1 and bag3 are similar (2675hz), about which there is no further study. (The possibility of material influence should be eliminated as much as possible)

5. The bags are filled with the same gas, have you ever thought about changing the filling, and observe the frequency change law?

the Quality of English Language is well.

Author Response

(The authors gave the same response as above.)

Reviewer 3 Report

  1. What are the components used in the test setup for structural dynamic measurements of the prosthesis?

  2. How were the boundary conditions (BCs) ensured for the test structure in terms of support?

  3. How many different weights were used during the calibration process, and what was the range of their masses?

  4. What is the significance of the major antiresonance observed at approximately 350 Hz in the measured data?

  5. Are the changes in resonance frequencies consistent across all investigated variants and bags? If not, what observations can be made regarding the sensitivity of certain
  6. Do the frequency values generally increase or decrease with increasing pressure for bag 1, according to the data?

Good

Author Response

(The authors gave the same response as above.)

Reviewer 4 Report

The authors have worked on "Approach for a non-intrusive detection of the fit of orthopaedic devices based on vibrational data". Following comments must be addressed before publishing in the Sensors journal. 

1. The vibration signal data and its processing details are not comprehensively discussed. Please use schematic flow chart to facilitate readers. Revise 2.3 with flow chart, showing input output signal and post processing approaches. 

2. How the reliability of experimental data is ensured? What measures are taken in this regard

3. Authors followed linear regression. Please highlight the analysis of variance or pca of data to signify the regression model

4. Include standard deviations in figure 5

5. Along with descriptive conclusions, point based highlights of critical take aways are recommended to be added. 

Please check grammatical errors throughout. 

Author Response

(The authors gave the same response as above.)

Round 2

Reviewer 2 Report

The authors have revised all the problems, and this paper can be published.

Reviewer 4 Report

The paper is improved.

Moderate changes required